# A Preliminary Survey of Olive Grove in Biskra (Southeast Algeria) Reveals a High Diversity of Thrips and New Records

**DOI:** 10.3390/insects13050397

**Published:** 2022-04-19

**Authors:** Chahrazed Warda Halimi, Malik Laamari, Arturo Goldarazena

**Affiliations:** 1Department of Nature and Life Sciences, University Mohamed Khider, BP 145 RP, Biskra 07000, Algeria; 2Laboratoire d’Amélioration des Techniques de Protection Phytosanitaires en Agro-Système Montagneux, University of Batna 1 Hadj Lakhdar, Batna 05000, Algeria; laamarimalik@yahoo.fr; 3Department of Agriculture, University of Batna 1 Hadj Lakhdar, Batna 05000, Algeria; 4Laboratorio Nacional de Referencia de Nemátodos y Artrópodos, Departamento de Biodiversidad y Ecologia Evolutiva, Museo Nacional de Ciencias Naturales, Despacho 411-CPle, Calle Serrano 115 Duplicado, 28006 Madrid, Spain; agoldaracena@mncn.csic.es

**Keywords:** *Neohydatothrips amygdali*, thrips, seasonal abundance, phenology of olive trees, Biskra

## Abstract

**Simple Summary:**

The diversity of thrips from olive trees in Algeria is unknown. According to the Ministry of Agriculture, thrip species have caused economic damage to the olive yield in the region of Biskra throughout the history of this crop, although the thrips pest species was unknown. To solve this question, we studied the thrips composition of one olive orchard over three years and the seasonal abundance of the species responsible of the damage of the olives in relation to the phenologic development of the olive tree

**Abstract:**

This study was conducted to determine Thysanoptera species composition associated with olive trees, fruit injury, fluctuations in the abundance of the most phytophagous species relative to the phenological stages of the olive tree, and the temporal variations in the species *Neohydatothrips*
*amygdali* in relation to environmental variations in Biskra province between 2018 and 2020. The olive orchard chosen for this study is located at the Experimental Station of the Technical Institute for the Development of Saharan Agriculture (ITDAS) in El Outaya (Biskra, a Saharan region of Algeria). Five trees were sampled each week, and thrips were collected by shaking 20 twigs (4 twigs per tree). Identification of thrips species was based on morphological characters of adults. Microscopic observation of the collected thrips specimens made it possible to report nine thrips species. Among them, four are reported for the first time in Algeria. Three taxa, namely *Haplothrips tritici*, *Neohydatothrips amygdali*, and *Frankliniella occidentalis* constituted the eudominant groups, representing 28.65%, 27.98%, and 23.39% of total specimens collected during the study, respectively. Thrips injury appeared as scaring and silvering of the fruit. The most abundant species was *H. tritici*, but *N. amygdali* was most common in the last two years of the survey. The influence of phenological stages of olive trees on the total number of *H. tritici* and *F. occidentalis* was significant. The highest numbers were recorded in flowering stage. The highest number of *N. amygdali* had two peaks: April (2020) and May (2018, 2019), as well as September (2020) and October (2018, 2019), when temperatures fluctuated between 20 °C and 30 °C. *N. amygdali* was totally absent during the other months (in winter and summer). During the three years of investigation, *N. amygdali* reproduced by thelytoky, with only females present.

## 1. Introduction

In Algeria, the olive tree is a major element of the agricultural economy. Its geographical position in the Mediterranean basin offers it an ideal climate for better quantitative and qualitative production. With more than 35 million olive trees, Algeria produces an average of one million tons of olives per year [1]. In the Saharan region, Biskra province occupies the first rank, with an established area of 4.501 ha (33% of the total) [2].

Several insects and pathogens affect olive trees, threatening olive production. Both the number of pests and diseases and their frequency have increased dramatically in recent years, causing serious damage to olive production [3]. *Liothrips oleae* (Costa) has been cited among olive pests in Spain [4], and it is distributed in France, Italy, Canary Island, former Yugoslavia, and Yemen [5]. The few studies that have been conducted to investigate thrips pests in olive groves are those of Canale et al. [6] in Italy, Rei et al. [7] in Portugal, Trdan et al. [8,9] in Slovenia, Agami et al. [10] in Egypt, and Marullo and Vono [11] in Italy. Despite this, very little is known about thrips fauna associated with olive tree.

Thysanoptera, commonly called thrips, are tiny and slender insects with fringed wings [12]. They have piercing and sucking-type mouths. They may can phytophagous, mycophagous, or predators of other arthropods [13,14,15]. Thrips are difficult to detect in crops due to their small size, mobility, and presence in tight and close places. Several species are of economic importance, mainly due to their polyphagous nature and ability to transmit plant pathogens that cause serious crop losses around the world [16,17]. They also have the ability to develop insecticide resistance. All these attributes contribute to the success of this phytophagous, which causes significant yield losses in agricultural systems. In contrast, some species of thrips are very useful and act as pollinators [15]. They also act as biological control agents [17,18,19,20]. Only a few surveys have been conducted in Algeria, and very little is known about Thysanoptera compared with other parts of the world.

The main objectives of this study were to determine the thrips species composition occurring on olive trees, as the species may be harmful to olive trees, and to study their population fluctuations in relationship to the phenology of olive tree. Seasonal abundance, affected by temperature and sex ratio, were studied only for *N. amygdali,* which was found to be the most abundant species in the last two years of the survey.

## 2. Materials and Methods

### 2.1. Study Site

The olive orchard chosen for this study was located at the Experimental Station of the Technical Institute for the Development of Saharan Agriculture (ITDAS) in El Outaya, 12 km north of Biskra (34°93′30.55″ N, 5°65′88.43″ E, 207 m). This region is characterized by a Saharan climate that is hot and dry in summer and cold in winter.

The studied olive grove, installed in 2005, covers an area of 40 hectares. It is planted in three plots: the first plot includes 345 trees (4 × 4 m), the second plot comprises 720 trees (4 × 2 m), and the last plot includes 812 trees (4 × 4 m). The irrigation system adopted is a drip system. The fertilizer used is urea (46% nitrogen). It was added in doses of 500 gr per tree before the budburst stage. During the winter rest, olive trees were pruned in order to maintain a correct balance between vegetation and fruiting and to obtain adequate ventilation and lighting. The olive grove is managed with an organic phytosanitary management system.

### 2.2. Thrips Community Composition and Thrips Attacks on Olive Trees

Monitoring was carried out for 3 years (2018–2019–2020). Samples were collected weekly using a shaking method of twigs (30 cm long) over a white board. During the survey, 5 olive trees were randomly selected from the same plot. From each tree, 4 twigs were shaken from the four cardinal points. All recovered thrips were removed from the tray using a fine camel-hair brush and placed in microtubes containing AGA solution (10 parts 60% ethanol, 1 part glacial acetic acid, and 1 part glycerol) [21,22]. The total number of thrips in larvae and adult stages was counted. Temperature was the only environmental variable studied (it rarely rains in this region, and relative humidity is generally low). Data were obtained from the meteorological station at the study site. A subsample of 50 flowers was dissected for observation of feeding injury to ovaries. Thrips feeding injuries were observed by larvae in late bloom; moreover, we observed scare punctuations from the first stage of fruit growth.

### 2.3. Mounting and Identification

Once in the laboratory, the thrips samples stored in AGA medium were transferred to a solution of NaOH (5%) for 12 h. Then, the specimens were transferred to an alcohol solution (60%) for 24 h. The thrips were mounted in Canada balsam [5]. The slides were examined under an optical microscope at magnifications of 10, 40 and 60, as identified in the scientific literature [23,24]. The larvae of thrips were also identified according to [24]. Voucher specimens were deposited in the insect collection of LATPPAM Research Laboratory, University of Batna.

### 2.4. Analyzing Abundance and Population Fluctuation of Thrips on Olive Trees

The relative abundance (*RA*) of species was calculated using the following equation [25]:
$$RAx\;=\;\frac{\mathrm{abundant}\;\mathrm{value}\;\mathrm{for}\;\mathrm{species}\;\times 100}{\mathrm{sum}\;\mathrm{of}\;\mathrm{abundance}\;\mathrm{values}\;\mathrm{for}\;\mathrm{all}\;\mathrm{species}}\;$$

After calculating relative abundance, species composition was determined according to Kucharczyk et al. [26], with some modifications. Species were divided into the following groups: eudominants (>15%), dominants (15–10.1%), subdominants (10–5.1%), recedents (5–1%), and subrecedents (<1%). For species of the eudominant group, the seasonal abundance was calculated for each year of the study. The thrips population abundance was analyzed according to the phenological stages of olive trees.

The study of population fluctuation of *N. amygdali* was more detailed because this species had a considerable presence during the three years of study and, which is herein reported for the first time in Algeria and on olive trees.

### 2.5. Data Analysis

Statical tests were carried out in order to demonstrate the influence of the phenological stage factor of the plant on the presence and abundance of thrips species and specify which stage is favorable for each species.

Statistical analysis was performed using the thrips densities obtained during the study (3 years). The data were subjected to analysis of variance, and means were separated by Tukey’s honestly significant difference (HSD) test at *p* < 0.05. All analyses were performed using Microsoft Statistics Program SPSS version 22 [27].

## 3. Results and Discussion

### 3.1. Thrips Community Composition and Thrips Attacks on Olive Trees

Results revealed the presence of nine thrips species recorded on olive trees during three years of investigation. These species belonging to four families, derived from two main suborders (Terebrantia and Tubulifera). Among these species, four are recorded for the first time in Algeria. This study revealed the presence of four thrips species for the first time on olive trees worldwide (Table 1).

In a previous study by Canal et al. [6], *A. collaris*, *A. intermedius*, and *H. andresi* were recorded in an olive grove in Tuscany, Italy. In the same country, Marullo and Vono [11] devoted a study to thrips attacking olive trees. Their study revealed the presence of only *Liothrips oleae*, the most common species on olive trees. Despite the economic importance of this species on olive trees, it was totally absent during the three years of investigation.

This study shows that thrips cause injury to olive fruit. Injury was apparent as scarring of the fruit surface. Fruit injury by thrips was not difficult to establish. Scars on the surface of the fruit are caused by oviposition and feeding activity of thrips. Silvering was the second type of injury, which occurs as a result of a sucking action of thrips extracting the content of plant cells, in particular the pigments they contain. As plant cells are drained of contents, they become filled with air. Injury results in the loss of the original color and the acquisition of the characteristic silvery appearance of injured olives. Figure 1 shows some of the most common types of physical symptoms observed on olive fruits. Only two studies have been conducted on thrips attacks on olive trees, which were conducted by Marullo and Vono [11] and Vono et al. [28] in Italy. The authors observed the same symptoms on olive fruits.

### 3.2. Analyzing Abundance of Thrips on Olive Trees

During the three years of the study, 4 318 specimens of the 9 species were collected from samples of olive trees: 957, 1588, and 1773 in 2018, 2019, and 2020, respectively. Three taxa—*H. tritici*, *N. amygdali*, and *F. occidentalis*—constituted the groups of eudominants, representing 28.65%, 27.98%, and 23.39% of total collected specimens, respectively. *N. amygdali* was collected more often in 2019 and 2020, representing 32.18% and 29.22% of total specimens, respectively. *H. tritici* was more abundant in 2018 than the other eudominant species (33.44%). This species breeds in Poaceae plants that grow in the olive orchards and their presence in the trees must be considered circumstantial. Additionally, *H. andresi* was the most abundant of the dominant species (13.59%), while *A. collaris* (1.27%) and *T. tabaci* (1.13%) were most abundant of the recedent species, and *M. fuscus* was the most abundant of the subrecedent species (0.30%) 2018 (Table 2).

### 3.3. Fluctuations in Numbers According to Phenological Stages

The obtained results show that activity of *N. amygdali* depends on the phenological stages of the olive tree (Figure 2). The first individuals were observed only after fruit formation (fruit set), which occurs before the end of April and the beginning of May. During 2018, this activity was only noticed after fruit enlargement (first stage of fruit growth). The petals of the flowers of the olive tree are not oviposition sites for *N. amygdali*, unlike many other species, in particular, *Frankliniella bispinosa* [29] or *F. occidentalis* [30]. Moreover, the absence of *N. amygdali* on the olive tree at the flowering stage confirms that it does not feed on pollen. Another period of intense activity was observed once the olives reached their maximum size (veraison). Similar results were reported by Hazir and Ulusoy [31], who recorded two peaks of thrips on nectarine, the first coinciding with the flowering stage (April) and the second, much larger peak at the fruit-ripening stage in May. In this study, a total absence of *N. amygdali* was noticed on the herbaceous plants present in the olive grove.

The abundance of *F. occidentalis* and *H. tritici* in olive trees varied depending on the phenological stage of the host plant. During the period at the beginning of the study, the number of species was relatively low due to the absence of flowers on the olive trees. However, after the bloom period, the numbers considerably increased (Figure 3 and Figure 4). Unlike *N. amygdali,* these two species were present in a single peak in April and the beginning of May. This peak coincides with the flowering stage. Our results are in agreement with those obtained by Canal et al. [6] and Hanafi et al. [32], who mentioned that the most abundant thrips populations occurred during flowering and warmer periods.

Results of statical tests showed that there was a significant effect of the phenological stage of the olive tree on the total number of *H. tritici* (F = 23.71, df = 17, *p* < 0.05). The phenological stages formed two homogeneous groups. The flowering stage was the most attractive for *H. tritici*, with an average of 257.33 individuals (Figure 5).

The same results were obtained with *F. occidentalis* (F = 7.73, df = 17, *p* < 0.05); the average number of this species during the flowering stage (171.33) was significantly higher than during the other phenological stages (Figure 6).

In contrast to the previous results, the studied olive phenological stages had no significant effect on the total number of *N. amygdali* (F = 1.65, df = 11, *p* > 0.05) during its presence on olive trees. All stages were in the same statistical group (Figure 7).

Dissection of the flowers allowed us to verify injury caused by thrips attacks. Given their peak abundance coinciding with flowering, *H. tritici* and *F. occidentalis* were responsible for some injuries observed on ovaries of dissected flowers and, later, on the fruits. *N. amygdali* likely contributed to additional injury as the ovaries matured into fruit due to its presence during the phenological stages of fruit growth and veraison.

Results of this study are in agreement with those reported by Gerin et al. [30], who stated that the most important factor that affects thrips population levels is the phenological stage of the plant. This is because many thrips species prefer flowers and young buds.

### 3.4. Temporal Fluctuation of Neohydatothrips Amygdali Based on Environmental Conditions

Monitoring of the temporal evolution of the numbers of *N. amygdali* revealed significant variations during the three years of the study. The most activity was recorded in May (spring) and October (autumn). The spring activity seems to be more important. In addition, it is noted that in 2020, thrips appeared on olive trees earlier than in other years (Figure 8).

According to Bournier [33], the evolution of thrips populations on crops depends on environmental conditions, in particular, temperature. However, thrips seem to be less abundant in the rainy season and more numerous during the dry season [34]. During the three years of study, the average monthly temperatures recorded during the months of January, February, and March did not exceed 17.9 °C. These temperatures may not have allowed this thrips species to regain its activity. Bournier [33], as well as Loomans and van Lenteren [35], mentioned that the temperatures favorable to the development of thrips are those between 25 and 30 °C. Lewis [17], Kucharczyk et al. [26], Bereś et al. [36], and Huruj et al. [37] stated that polyphagous thrips species show two peaks of abundance in their seasonal dynamics in temperate climates. The first peak occurs in spring and is caused mainly by wintering adult individuals colonizing crops. These authors added that according to the weather conditions, at the end of May, after laying eggs, the population of adults decreases. Indeed, the temperatures recorded during the months of April and May [20 °C–30 °C], allowed *N. amygdali* to record a first peak of development.

This first peak of abundance is formed by both the larvae and adults of a new generation (Figure 9). The comparative number of larvae on 16th April was significantly higher in 2020 compared to 2019; the number was less in 2018. During this period, peaks were observed in April for 2018–2020 and later in May for 2019. In the province of Fars (Iran), which is also a Saharan region, Minaei [38] reported significant activity of *N. amygdali* during the month of May. In Riadh and Baha (Saudi Arabia), Rassool et al. [39] reported a first development peak of *N. amygdali* during April in 2019. These two regions are also characterized by a desert climate that is dry and very hot in summer, similar to the climate of our study region. During the months of June, July, and August, thrips completely disappeared. It is possible that the average monthly temperatures recorded during this period (maximum of 37.1 °C in July 2018) did not allow thrips to continue their development. Thrips populations disperse or die out when conditions become unfavorable. Lewis [17], Reitz [40], and Zhang et al. [41] confirmed that temperatures above 30 °C cause the death of larvae and pupae. Dispersing populations colonize the crop in large numbers when conditions become favorable. Thrips reappeared in September (2020) and in October (2018 and 2019), when temperatures were very close to 25 °C. No more immature specimens were found on olive fruit during this period.

It is remarkable that *Liothrips oleae*, the species that caused economic losses in the production of olive trees during the 18th century in Mediterranean areas, has not been found in this area of Algeria.

### 3.5. Sex Ratio of Neohydatothrips amygdali

During the three years of study, the populations of *N. amygdali* collected on olive trees in the region of Biskra were exclusively female. These results are consistent with those obtained by Minaei [38] in Iran on *Amygdalus scoparia* [Rosales: Rosaceae]. However, Rassool et al. [39] collected both males and females of *N. amygdali* in Saudi Arabia on *Acacia seyal* [Fabales: Fabaceae] and *A. ehrenbergiana* [Fabales: Fabaceae]. The sex-ratio index is very important in the study of the biotic potential of a population. It is also important for understanding the causes and consequences in the differences in population structures and mating systems [42]. Predomination of females over males has consequences for individuals, populations, and species, allowing females to transmit their successful genotypes to all of their offspring; produce only daughters, maximizing the rate of increase; and eliminate the need to find or to attract a mate. In contrast, sexual reproduction results in offspring with a diversity of genotypes; the production of males, which themselves cannot produce offspring; and an inability to reproduce without males being locally present and without diverting some amount of time and energy to the mating process [43]. Thysanoptera exhibit the three types of parthenogenesis: arrhenotoky, thelytoky, and deuterotoky [17,18,19,20,21,22,23,24,25,26,27,28,29,30,31,32,33,34,35,36,37,38,39,40,41,42,43,44]. The mode of sex determination is haplodiploidy, which means females are diploid and males are always haploid [45,46]. Most species populations are bisexual with predominance of females. In some species, males are rare or unknown. Some species have confirmed their reproduction by thelytoky, such as *Heliothrips haemorrhoidalis, H. errans, Scirtothrips longipennis,* and *Leucothrips nigripennis*. Males of these species are very rare, if ever recorded [17]. Stouthamer et al. [47] and Arakaki et al. [48] confirmed that the endosymbiotic bacterium of the genus *Wolbachia* is responsible for this reversible thelytoky, and they reported that the elimination of this bacterium by antibiotics or by high temperature induces the production of males. This was recently been proven by Kumm and Moritz [49]. In other thrips species, despite the absence of this bacteria, they are unable to produce males. This is the case of *H. haemorrhoidalis* [50].

## 4. Conclusions

Thrips species were represented by nine species in an olive grove in Biskra region during 2018, 2019, and 2020. Among them, four species were recorded for the first time in Algeria: *A. collaris, F. megalops*, and *H. andresi* as predator, as well as *N. amygdali* as phytophagous. Relative abundance allows us to classify these species in five groups. The most abundant species, constituting the eudominant group, wer: *H. tritici*, *N. amygdali*, and *F. occidentalis.* Phenological stages of the olive tree are likely to be an important factor affecting the population dynamics of *H. tritici, N. amygdali*, and *F. occidentalis*. Higher abundances of *H. tritici* and *F. occidentalis* were observed in the flowering stage of olive trees. Although this stage did not attract *N. amygdali*, the fruit growth stage caused its appearance and abundance. A reappearance of *N. amygdali* was recorded when the fruit reached full development. Monitoring of the seasonal abundance of *N. amygdali* on olive trees revealed significant variations. Two population peaks were observed, with infestation of the olive trees beginning in April, with important activity in May (spring) and October (autumn). Thrips were present in the field when temperatures were close to 25 °C, but no presence of thrips was recorded when temperatures rose above 30 °C. The presence of frequent and numerous *N. amygdali* immature stages confirmed olive trees as important host plants for this species. Finally, all populations of *N. amygdali* collected were represented only by females. Further studies should focus on the reason for the absence of males of *N.amygdali* on olive trees. According to the data and field observations obtained during this study, the presence of phytophagous thrips caused injury to olive fruits, but additional tests and statistical analysis are required to determine the species responsible for the injury and to establish a correlation between the abundance of thrips and their level of damage.

## Figures and Tables

**Figure 1 insects-13-00397-f001:**
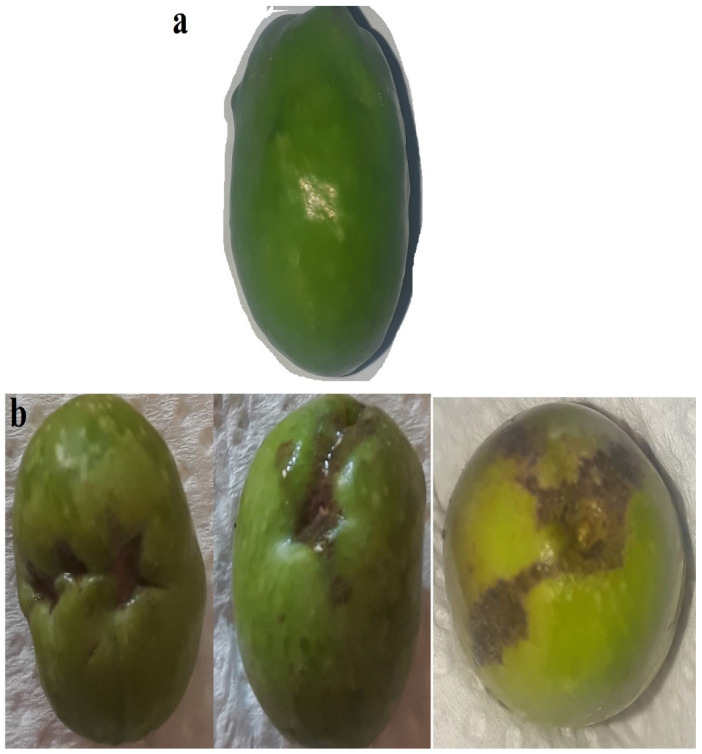
(**a**) Undamaged olive; (**b**) olive fruit injury.

**Figure 2 insects-13-00397-f002:**
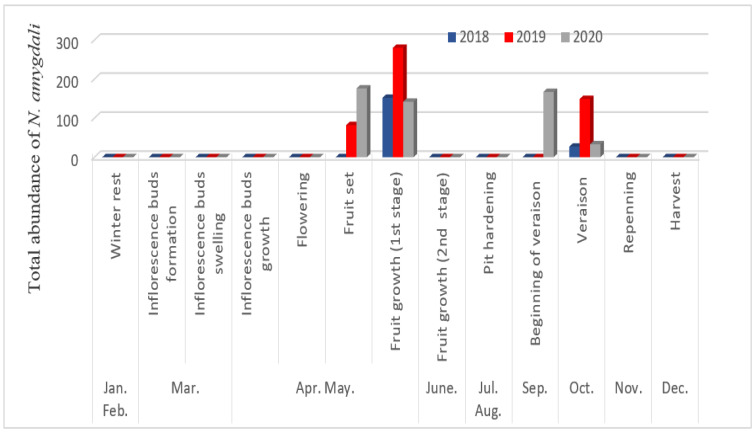
Period of activity of *Neohydatothrips amygdali* according to the phenological stages of growth of olive trees in Biskra region.

**Figure 3 insects-13-00397-f003:**
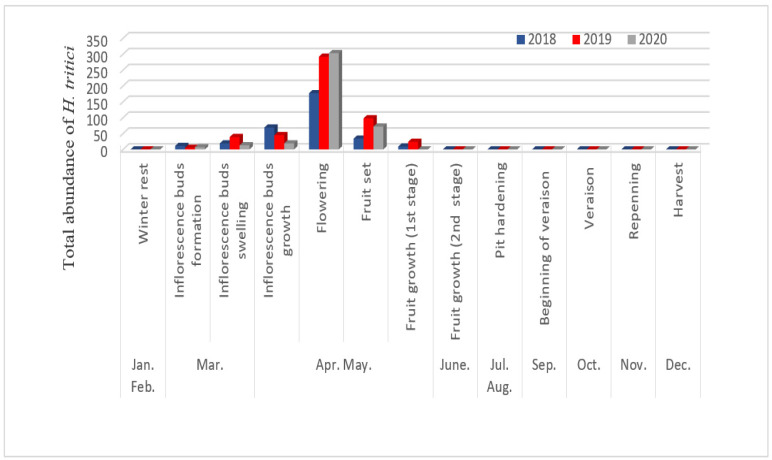
Period of activity of *Haplothrips tritici* according to the phenological growth stage of olive trees in Biskra region.

**Figure 4 insects-13-00397-f004:**
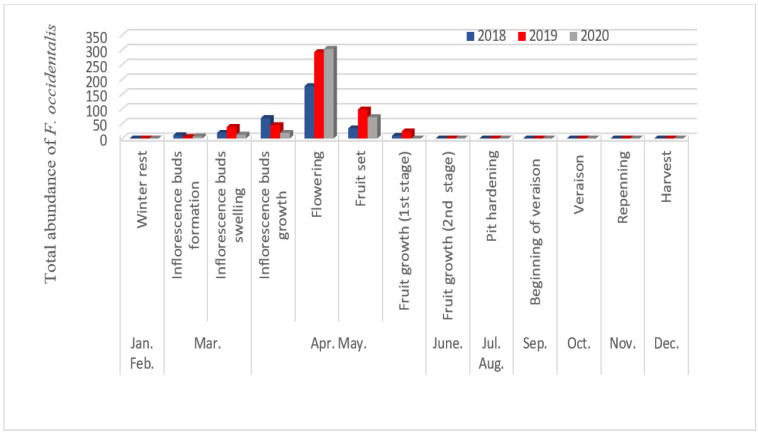
Period of activity of *Frankliniella occidentalis* according to the phenological growth stage of olive trees in Biskra region.

**Figure 5 insects-13-00397-f005:**
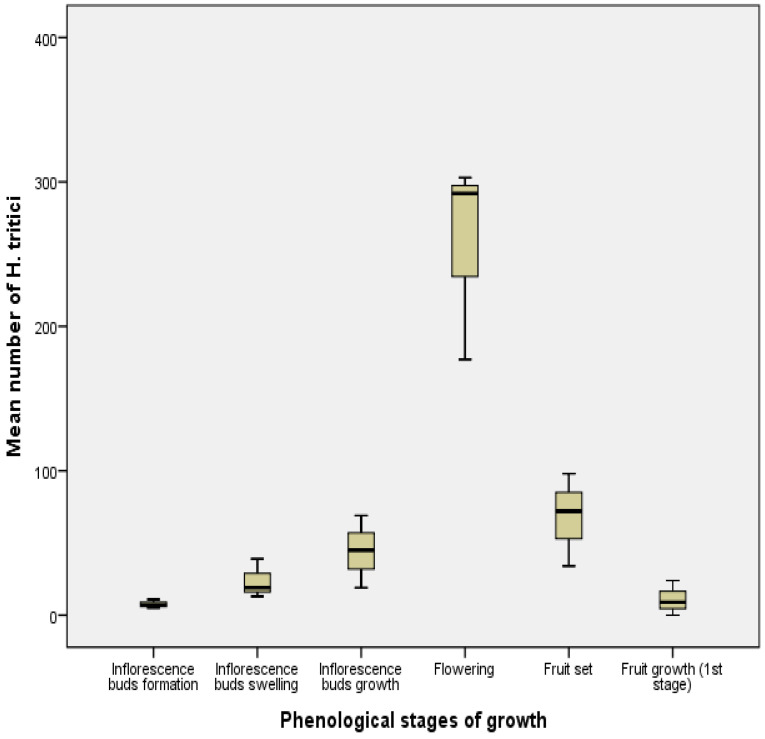
Mean number (±SE) of *Haplothrips tritici* during different phenological stages of olive tree growth in Biskra region.

**Figure 6 insects-13-00397-f006:**
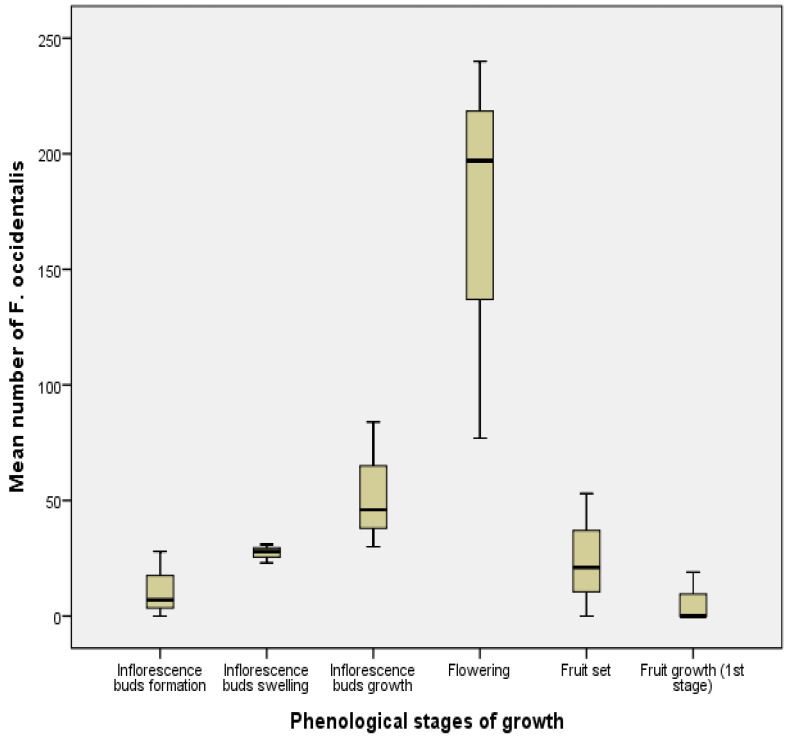
Mean number (±SE) of *Frankliniella occidentalis* during different phenological stages of olive tree growth in Biskra region.

**Figure 7 insects-13-00397-f007:**
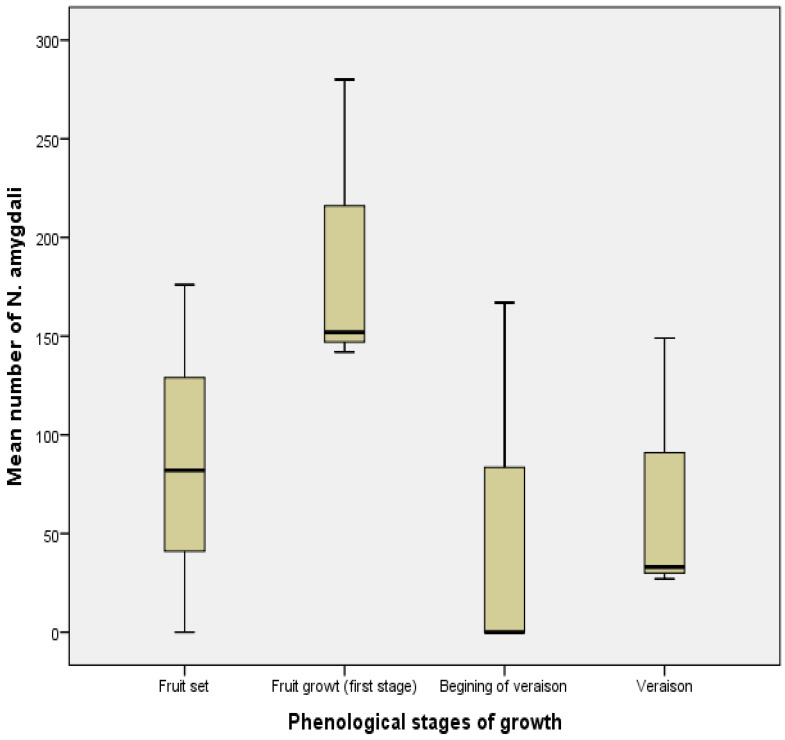
Mean number (±SE) of *Neohydatothrips amygdali* during different phenological stages of olive tree growth in Biskra region.

**Figure 8 insects-13-00397-f008:**
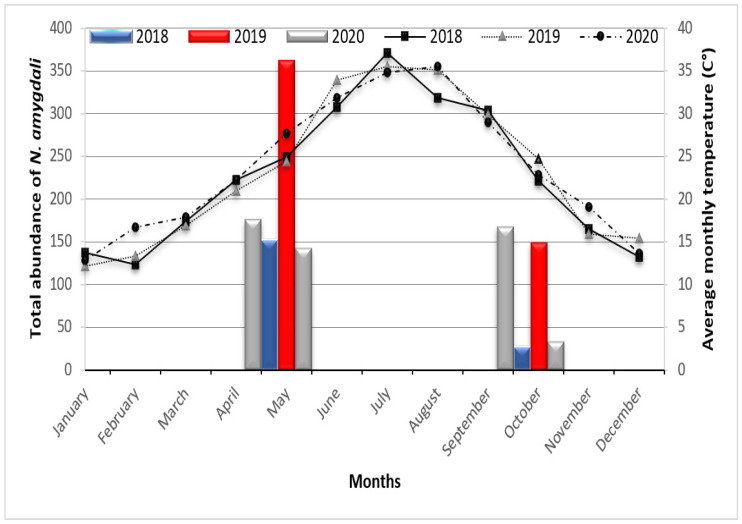
Temporal fluctuation of *Neohydatothrips amygdali* adults on olive trees in Biskra region according to monthly average temperatures.

**Figure 9 insects-13-00397-f009:**
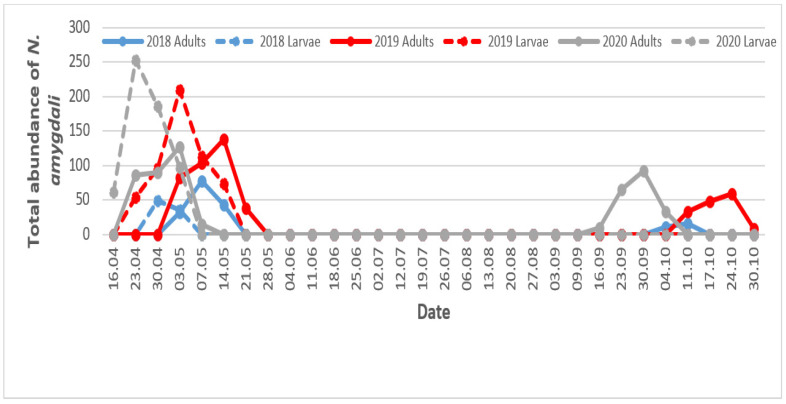
Seasonal abundance of *Neohydatothrips amygdali* adults and larvae on olive trees in Biskra region.

**Table 1 insects-13-00397-t001:** Thrips species composition in an olive grove in Biskra region.

Suborder	Family	Species
Terebrantia		* *Aeolothrips collaris* Priesner, 1919
	Aeolothripidae	*Aeolothrips intermedius* Bagnall, 1934
		*° *Franklinothrips megalops* (Trybom, 1912)
	Melanthripidae	° *Melanthrips fuscus* Sulzer, 1776
		*Frankliniella occidentalis* (Pergande, 1895)
	Thripidae	*° *Neohydatothrips amygdali* Minaei, 2016
		*Thrips tabaci* Lindeman, 1889
Tubulifera	Phlaeothripidae	* *Haplothrips andresi* Priesner, 1931
		° *Haplothrips tritici* (Kurdjumov, 1912)

(*) first record in Algeria, (°) first record on olive tree.

**Table 2 insects-13-00397-t002:** Thrips species composition and relative abundance in an olive grove in Biskra region.

Species	2018		2019		2020		Total of Three Years	
	Value	*RA*%	Value	*RA*%	Value	*RA*%	Value	*RA*%
(●) *Aeolothrips collaris*	10	1.04%	8	0.50%	37	2.09%	55	1.27%
(●) *Aeolothrips intermedius*	4	0.42%	1	0.06%	20	1.13%	25	0.58%
(◊) *Franklinothrips megalops*	4	0.42%	17	1.07%	113	6.37%	134	3.10%
(*) *Melanthrips fuscus*	6	0.63%	4	0.25%	3	0.17%	13	0.30%
(*) *Frankliniella occidentalis*	216	22.57%	380	23.93%	414	23.35%	1010	23.39%
(*) *Neohydatothrips amygdali*	179	18.70%	511	32.18%	518	29.22%	1208	27.98%
(*) *Thrips tabaci*	29	3.03%	20	1.26%	0	0.00%	49	1.13%
(◊) *Haplothrips andresi*	189	19.75%	144	9.07%	254	14.33%	587	13.59%
(*) *Haplothrips tritici*	320	33.44%	503	31.68%	414	23.35%	1237	28.65%
Total	957	100.00%	1588	100.00%	1773	100.00%	4318	100.00%

*RA* = relative abundance, (●) facultative predator, (◊) obligatory predator, (*) phytophagous.

## Data Availability

Data is contained within the article.

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
