# Peer review of "A Preliminary Survey of Olive Grove in Biskra (Southeast Algeria) Reveals a High Diversity of Thrips and New Records"

_insects, 2022, doi:10.3390/insects13050397_

Round 1
Reviewer 1 Report
The article, "Seasonal Abundance of Neohydatothrips amygdali (Minaei,
2016) (Thysanoptera: Thripidae) on Olive Tree Orchard in
South East Algeria. " offers new information on the species composition and seasonal abundance of thrips found on olive trees in South East Algeria. This article is very succinct and easy to read, getting right to the point with no superfluous information. This efficiency is a strength of the article, however there are some weaknesses. The methods section needs more information and there are some inconsistencies with the language throughout the text.
In the methods section it is stated that branches were shaken weekly, 20 twigs from 5 trees. This needs further details. How many branches per tree were shaken, 4 twigs on each tree, or 20 twigs on each of the 5 trees for a total of 100? This information is inconsistent in the abstract as well. How were the trees and twigs chosen for sampling (random, haphazard?), and were they all in the same plot or dispersed throughout the different plots? What were the branches shaken over, a sheet, a white board, paper? And how were the thrips then collected, was it the paintbrush method, an aspirator or other method?
Were the larvae also identified to species?
In the results section it would be helpful to have a written description of the damage to the olive fruits in addition to the picture.
There are a few issues with the spelling and grammar, but two repeated errors are the use of "phonological" instead of phenological in many places in the text and on figures, that should be corrected. The other repeated error is on figures 8 and 9 where it says "effectif", what is this? It is not an English word.
This article provides important information on the species composition of thrips on olive trees, but it needs some additional details in the methods section and more information regarding thrips damage to olive fruits.
Author Response
Reviewer 1
Comments and Suggestions for Authors
The article, "Seasonal Abundance of Neohydatothrips amygdali (Minaei,
2016) (Thysanoptera: Thripidae) on Olive Tree Orchard in
South East Algeria. " offers new information on the species composition and seasonal abundance of thrips found on olive trees in South East Algeria. This article is very succinct and easy to read, getting right to the point with no superfluous information. This efficiency is a strength of the article, however there are some weaknesses. The methods section needs more information and there are some inconsistencies with the language throughout the text.
In the methods section it is stated that branches were shaken weekly, 20 twigs from 5 trees. This needs further details.
How many branches per tree were shaken, 4 twigs on each tree, or 20 twigs on each of the 5 trees for a total of 100?
Answer: 5 trees were randomly selected, 4 twigs were shaken from each tree for a total of 20 twigs per week. This is added now in line 90-92.
This information is inconsistent in the abstract as well. How were the trees and twigs chosen for sampling (random, haphazard?),and were they all in the same plot or dispersed throughout the different plots?
Answer: The trees and twigs were chosen randomly in the same plot. This is added now in line 90.
What were the branches shaken over, a sheet, a white board, paper?
Answer: Branches were shaken over a white board. This is added now in line 89.
And how were the thrips then collected, was it the paintbrush method, an aspirator or other method?
Answer: Thrips were removed using a fine camel-hair brush. This is added now in line 92.
Were the larvae also identified to species?
Answer: Yes, the larvae of Neohydatothrips were identified following Vierbergen et al., 2010. This is added in line 108.
In the results section it would be helpful to have a written description of the damage to the olive fruits in addition to the picture.
Answer: Description added now in line 148-154.
There are a few issues with the spelling and grammar, but two repeated errors are the use of "phonological" instead of phenological in many places in the text and on figures, that should be corrected.
Answer: This is corrected all the text.
The other repeated error is on figures 8 and 9 where it says "effectif", what is this? It is not an English word.
Answer: This is corrected now in all figures.
This article provides important information on the species composition of thrips on olive trees, but it needs some additional details in the methods section and more information regarding thrips damage to olive fruits.
Reviewer 2 Report
This study aimed to fill the knowledge gap of thrips diversity on olive trees in Algeria by conducting surveys of thrips from one orchard in Biskra. Thrips were collected from five trees within the grove over the course of 3 y from 2018 to 2020. Specimens were identified using morphological features and species were grouped into categories based on relative abundance. Nine thrips species were identified, with four reported for the first time in Algeria. Emphasis was placed on Neohydatothrips amygdali as the most significant finding, as it has been reported here on olive for the first time. It was implicated in causing damage to the fruit and its abundance has two peaks coinciding with fruit set and ripening. A strength of this paper is the analysis of thrips abundance with respect to plant phenology, which can help to better understand thrips population dynamics in olive groves. Further analyses on seasonality with respect to plant phenology and impacts of temperature are discussed for N. amygdali and other abundant species collected.
General concept comments
My main concern is the implication of N. amygdali in causing the damage reported. While its abundance/presence is certainly tightly correlated with the presence of the onset of fruit, more evidence needs to be provided that N. amygdali is causing the majority of the damage seen. Could F. occidentalis and/or H. tritici feeding on the ovaries of flowers result in damage that becomes more apparent as the ovary expands and the fruit begins to set? Have any tests excluding N. amygdali from fruit after F. occidentalis and/or H. tritici have fed on flowers been conducted to see if damage appears? If you have direct evidence that N. amygdali caused the damage reported, please elaborate further as well as include literature on the biology of N. amygdali (if available).
The methods section should include more details on thrips collection methods. See specific comments below.
The conclusion section would benefit by re-emphasizing the thrips species that are newly documented on olive trees and new on olive trees in Algeria. Mention of the potential damage these new species can cause should also be included. Adding a concluding remark about the importance of monitoring for these species in olive groves could broaden the impacts and scope of this paper.
I did not make any corrections for spelling, formatting, or English language. The text will require extensive editing of English langue and style.
Specific comments
Title: while Neohydatothrips amygdali was a major focus of the paper, the title should also incorporate some aspect of quantifying thrips diversity, as this was the original objective. See comment on line 62 below.
Line 27: This does not match with the methods described at line 77, which states that there were 20 twigs sampled from five trees.
Lines 30-31: The order of abundance does not match that in Table 2. It should be Haplothrips tritici, Neohydatothrips amygdali, then Frankliniella occidentalis.
Line 32: Haplothrips tritici was most abundant overall. Maybe clarify that Neohydatothrips amygdali was most common in the last 2 years of the survey.
Line 51: Instead of “in this way”, I would suggest “to investigate thrips pests in olive groves” to clarify what the literature is referring to.
Lines 60-61: Please include the (Order: Family) for this plant after the species.
Line 62: Another main component of the study was to survey olive trees for all potential thrips species, and this should be mentioned in this section.
Line 74: Provide a sentence or two explaining the management (e.g., irrigation, fertilizer, and pest management practices like the use of organic pesticides, trap crops, mulch, biological control, etc.).
Line 77: There should be more details on the collection method. Were the twigs that were shaken left attached to the trees? Were the same twigs shaken each time or were new twigs randomly selected each time? Were the same 5 trees surveyed each time? How were thrips collected after being shaken off the twigs? Did thrips fall onto a white sheet held under the twigs? Were they aspirated off a sheet or shaken directly into a collection container? Please clarify whether 20 twigs were sampled from each of the five trees or 20 twigs in total each week, with four from each of the five trees.
Lines 83-84: Were any flower subsamples dissected to examine ovaries for feeding damage?
Line 109: add “suborders” after “main”.
Lines 119-123: This section refers to fruit damage clearly being caused by thrips with support from the literature based on damage caused by Liothrips olieae. I agree the damage depicted in Figure 1 is thrips damage but, without further evidence or experimentation, I don’t think it can be solely attributed to Neohydatothrips amygdali. I would recommend the caption for Figure 1 to read “Figure 1. Olives with scarring damage caused by Haplothrips tritici, Neohydatothrips amygdali, and/or Frankliniella occidentalis.”
Line 135: Species should be spelled out completely in the table. For the table legend, I would stick to using ° and * for indicating first record on olive trees and first record in Algeria, respectively.
Figure 1: Please include scale bars in each image and italicize species.
Figure 5 should be Figure 2. Please fix this and the subsequent figure numbering in both figure captions and in-text references.
Figures 5-7: Species should be fully spelled out. Keep the order of years consistently presented: 2018-2019-2020. I suggest adding another horizontal axis that depicts the time of year to each graph. Alternatively, a table could precede the graphs that shows an average timeline of plant phenological changes throughout the year.
Line 143: Please provide evidence for this. Were flower petals dissected/examined for eggs?
Line 168: I suggest “3.4. Temporal fluctuations of Neohydatothrips amygdali based on environmental conditions”
Lines 171-172: Include the reason the spring peak is most important.
Line 178: Figure 8 should be Figure 5.
Figure 8: Why are there two columns for 2020? Spell out species names completely in the figure. Y-axis should be total abundance?
Line 192: Figure 9 should be Figure 6.
Line 197: Please clarify whether Rassool et al. and other authors cited in the sentences that follow are referring to Neohydatothrips amygdali or that you are drawing conclusions based other species of thrips in the literature that are being regarded as surrogates for N. amygdali.
Figure 9: Spell out species names completely in the figure. Y-axis should be total abundance?
Line 212: Figure 10 should be Figure 7. Weren’t females were found in all three years?
Figure 10: Spell out species names completely in the figure.
Line 218: Please clarify whether Rassool et al. are referring to Neohydatothrips amygdali sex ratios, or ratios for other closely related thrips.
Lines 219-220: Please add brief mention of how sex ratios affect biotic performance.
Lines 239-247: I would list by order of abundance: Haplothrips tritici, Neohydatothrips amygdali, then Frankliniella occidentalis.
Lines 242-243: The positive correlation is for temperatures up to 25 C, but add the caveat that thrips are not present when temperatures rise above 30 C.
Author Response
Reviewer 2 This study aimed to fill the knowledge gap of thrips diversity on olive trees in Algeria by conducting surveys of thrips from one orchard in Biskra. Thrips were collected from five trees within the grove over the course of 3 y from 2018 to 2020. Specimens were identified using morphological features and species were grouped into categories based on relative abundance. Nine thrips species were identified, with four reported for the first time in Algeria. Emphasis was placed on Neohydatothrips amygdali as the most significant finding, as it has been reported here on olive for the first time. It was implicated in causing damage to the fruit and its abundance has two peaks coinciding with fruit set and ripening. A strength of this paper is the analysis of thrips abundance with respect to plant phenology, which can help to better understand thrips population dynamics in olive groves. Further analyses on seasonality with respect to plant phenology and impacts of temperature are discussed for N. amygdali and other abundant species collected.
General concept comments
My main concern is the implication of N. amygdali in causing the damage reported. While its abundance/presence is certainly tightly correlated with the presence of the onset of fruit, more evidence needs to be provided that N. amygdali is causing the majority of the damage seen.
Could F. occidentalis and/or H. tritici feeding on the ovaries of flowers result in damage that becomes more apparent as the ovary expands and the fruit begins to set? Have any tests excluding N. amygdali from fruit after F. occidentalis and/or H. tritici have fed on flowers been conducted to see if damage appears? If you have direct evidence that N. amygdali caused the damage reported, please elaborate further as well as include literature on the biology of N. amygdali (if available).
Answer: in this study, no test was done to prove which species is responsible for the damage to olives. We will do it in a future investigation. However the lower populations founded of Frankliniella occidentalis (polyphagous thrips) and Haplothrips tritici (that is specific from grass –Poaceae-) and the high populations of N. amygdali in the last two years of sampling could suggest that this species could be involved in the observed damage.
The methods section should include more details on thrips collection methods. See specific comments below.
Answer We have improved the description of the methodololgy according to the reviewers
The conclusion section would benefit by re-emphasizing the thrips species that are newly documented on olive trees and new on olive trees in Algeria.
Answer: this is added in line 307-209.
Mention of the potential damage these new species can cause should also be included.
Answer: this is added in line 325-326
Adding a concluding remark about the importance of monitoring for these species in olive groves could broaden the impacts and scope of this paper.
Answer: this is added in line 324-328
I did not make any corrections for spelling, formatting, or English language. The text will require extensive editing of English langue and style.
Specific comments
Title: while Neohydatothrips amygdali was a major focus of the paper, the title should also incorporate some aspect of quantifying thrips diversity, as this was the original objective. See comment on line 62 below.
Answer: this is added now in line 2
Line 27: This does not match with the methods described at line 77, which states that there were 20 twigs sampled from five trees.
Answer: this is added now in line 26-27.
Lines 30-31: The order of abundance does not match that in Table 2. It should be Haplothrips tritici, Neohydatothrips amygdali, then Frankliniella occidentalis.
Answer: this is added now in line 30.
Line 32: Haplothrips tritici was most abundant overall. Maybe clarify that Neohydatothrips amygdali was most common in the last 2 years of the survey.
Answer: this is added now in line 33.
Line 51: Instead of “in this way”, I would suggest “to investigate thrips pests in olive groves” to clarify what the literature is referring to.
Answer: this is added now in line 52.
Lines 60-61: Please include the (Order: Family) for this plant after the species.
Answer: this paragraph has been deleted.
Line 62: Another main component of the study was to survey olive trees for all potential thrips species, and this should be mentioned in this section.
Answer: this is added now in line 69-73.
Line 74: Provide a sentence or two explaining the management (e.g., irrigation, fertilizer, and pest management practices like the use of organic pesticides, trap crops, mulch, biological control, etc.).
Answer: this is added now in line 82-86.
Line 77: There should be more details on the collection method. Were the twigs that were shaken left attached to the trees? Were the same twigs shaken each time or were new twigs randomly selected each time? Were the same 5 trees surveyed each time? How were thrips collected after being shaken off the twigs? Did thrips fall onto a white sheet held under the twigs? Were they aspirated off a sheet or shaken directly into a collection container? Please clarify whether 20 twigs were sampled from each of the five trees or 20 twigs in total each week, with four from each of the five trees.
Answer: these are added now in line 89-92.
Lines 83-84: Were any flower subsamples dissected to examine ovaries for feeding damage?
Answer: Yes, it is added in Lines 97-100
Line 109: add “suborders” after “main”.
Answer: this is added now in line 140.
Lines 119-123: This section refers to fruit damage clearly being caused by thrips with support from the literature based on damage caused by Liothrips olieae. I agree the damage depicted in Figure 1 is thrips damage but, without further evidence or experimentation, I don’t think it can be solely attributed to Neohydatothrips amygdali. I would recommend the caption for Figure 1 to read “Figure 1. Olives with scarring damage caused by Haplothrips tritici, Neohydatothrips amygdali, and/or Frankliniella occidentalis.”
Answer: this is added now in line 158.
Line 135: Species should be spelled out completely in the table. For the table legend, I would stick to using ° and * for indicating first record on olive trees and first record in Algeria, respectively.
Answer: this is added now in line 169.
Figure 1: Please include scale bars in each image and italicize species.
Answer: this is added now in line 172.
Figure 5 should be Figure 2. Please fix this and the subsequent figure numbering in both figure captions and in-text references.
Answer: this is added now in line 187.
Figures 5-7: Species should be fully spelled out. Keep the order of years consistently presented: 2018-2019-2020. I suggest adding another horizontal axis that depicts the time of year to each graph. Alternatively, a table could precede the graphs that shows an average timeline of plant phenological changes throughout the year.
Answer: this is added now in Figures.
Line 143: Please provide evidence for this. Were flower petals dissected/examined for eggs?
Answer: this is added now in line 101.
Line 168: I suggest “3.4. Temporal fluctuations of Neohydatothrips amygdali based on environmental conditions”
Answer: this is added now in line 1228.
Lines 171-172: Include the reason the spring peak is most important.
Answer: this is added now in line 240.
Line 178: Figure 8 should be Figure 5.
Answer: this is corrected now.
Figure 8: Why are there two columns for 2020? Spell out species names completely in the figure. Y-axis should be total abundance?
Answer: because in this year, the presence of this species was in two successive months.
Line 192: Figure 9 should be Figure 6.
Answer: this is corrected now.
Line 197: Please clarify whether Rassool et al. and other authors cited in the sentences that follow are referring to Neohydatothrips amygdali or that you are drawing conclusions based other species of thrips in the literature that are being regarded as surrogates for N. amygdali.
Answer: this is added now in line 256.
Figure 9: Spell out species names completely in the figure. Y-axis should be total abundance?
Answer: this is added now in line 247.
Line 212: Figure 10 should be Figure 7. Weren’t females were found in all three years?
Answer: maybe this is a sign proving that olive tree is a favorable host which presents a good source of habitat, food and oviposition for this thrips.
Figure 10: Spell out species names completely in the figure.
Answer: reviewer 3 proposed to delete this figure of sex ratio. All information about this figure are include in the text.
Line 218: Please clarify whether Rassool et al. are referring to Neohydatothrips amygdali sex ratios, or ratios for other closely related thrips.
Answer: this is added now in line 283.
Lines 219-220: Please add brief mention of how sex ratios affect biotic performance.
Answer: when females predominate over males, this increases their dispersion of the species and their reproductive performance. this is added now in line 287-293.
Lines 239-247: I would list by order of abundance: Haplothrips tritici, Neohydatothrips amygdali, then Frankliniella occidentalis.
Answer: this is added now in line 311.
Lines 242-243: The positive correlation is for temperatures up to 25 C, but add the caveat that thrips are not present when temperatures rise above 30 C.
Answer: this is added now in line 321.
Reviewer 3 Report
#Manuscript Title: Seasonal Abundance of Neohydatothrips amygdali (Minaei, 2016) (Thysanoptera: Thripidae) on Olive Tree Orchard in Southeast Algeria.
The paper describes new species of thrips attacking olive crops in Algeria. It is really interesting, especially for a taxonomy group of thrips with new records in olive. But the paper presents some mismatches among title, aim, and data presentation. Plus, a statistical analysis is missing to support the author’s claim. Please see my comments and suggestion below. Then, my initial recommendation is a major review.
#Comments
L 1. The manuscript title does not match the paper research. The authors report species richness and abundance on olive crops; recorded new species attacking the crop for the first time and in Algeria. So, there are more findings behind the title. I suggest modifying it.
L 13-22. The sentence appears fragmented, and there is no conclusion. Please rewrite and include the paper conclusion.
L 26-27. If it was five twigs per tree, the total number should be 25. (Five trees sampled x 5 twigs per tree). Please verify.
L 31. Here and in the entire manuscript, the decimal format is . not , (like 28.85%).
L 39. Only two words are not presented in the title. I suggest they change the similar ones to increase the paper range.
L 55-65. It is unclear the authors’ aim. Here, it seems that it was related to a unique species. But, in the results, it is not the case. Please revise. Also, this species abundance was a finding of the research, not its aim.
L 90-92. Can the authors provide the accession numbers from the species deposited? It would be constructive for future researchers to compare them.
L 99. Provide a reference for such classification.
L 101. Correct the typo error ‘phonological’.
L 93-104. The authors solely descriptively report the results without any analysis to support the claims. I would suggest analyzing the data to provide a shred of better evidence. For instance, the authors can compare species abundance according to year phenological stage and associate the temperature with absolute counts. Plus, it was average air temperature? If so, please mention it.
L 121-123. These sentences are not connected, and it is unclear that symptoms in line 123 refer to common types (which means common here?) in 121. Besides, these symptoms can be described here.
L 124-125. Please provide detailed captions, mostly about damage symptoms. The authors might also consider including a healthy olive by the side to improve it.
L 135-136. Family mention can be removed here since it was already informed in Table 2.
L 138. The caption is incomplete.
L 162-167. Figure legends are in the incorrect order and also incomplete. Plus, as there were zero counts in some stages, it can be removed to improve figure presentation.
L 187-190. It appears that temperature might do not affect species abundance. But as I said previously, an analysis is necessary to do so.
L 212-214. The figure is unnecessary as the authors can include the information in the text.
L 224. Species presence was familiar under 20 °C (Figure 8); there was only in peak when the temperature reached 25 °C in 2019.
Author Response
Reviewer 3
The paper describes new species of thrips attacking olive crops in Algeria. It is really interesting, especially for a taxonomy group of thrips with new records in olive. But the paper presents some mismatches among title, aim, and data presentation. Plus, a statistical analysis is missing to support the author’s claim. Please see my comments and suggestion below. Then, my initial recommendation is a major review.
#Comments
L 1. The manuscript title does not match the paper research. The authors report species richness and abundance on olive crops; recorded new species attacking the crop for the first time and in Algeria. So, there are more findings behind the title. I suggest modifying it.
Answer: this is added now in line 2
L 13-22. The sentence appears fragmented, and there is no conclusion. Please rewrite and include the paper conclusion.
L 26-27. If it was five twigs per tree, the total number should be 25. (Five trees sampled x 5 twigs per tree). Please verify.
Answer: this is added now in line 26-27.
L 31. Here and in the entire manuscript, the decimal format is . not , (like 28.85%).
Answer: this is corrected
L 39. Only two words are not presented in the title. I suggest they change the similar ones to increase the paper range.
Answer: this is added now in line 40.
L 55-65. It is unclear the authors’ aim. Here, it seems that it was related to a unique species. But, in the results, it is not the case. Please revise. Also, this species abundance was a finding of the research, not its aim.
Answer: this is added now in line 69-73.
L 90-92. Can the authors provide the accession numbers from the species deposited? It would be constructive for future researchers to compare them.
Answer: this is added now in line 108.
L 99. Provide a reference for such classification.
Answer: this is added now in line 119
L 101. Correct the typo error ‘phonological’.
Answer: this is added now in line 118.
L 93-104. The authors solely descriptively report the results without any analysis to support the claims. I would suggest analyzing the data to provide a shred of better evidence. For instance, the authors can compare species abundance according to year phenological stage and associate the temperature with absolute counts. Plus, it was average air temperature? If so, please mention it.
Answer: this is added now in line 203-223
L 121-123. These sentences are not connected, and it is unclear that symptoms in line 123 refer to common types (which means common here?) in 121. Besides, these symptoms can be described here.
Answer: this is added now in line 139-144.
L 124-125. Please provide detailed captions, mostly about damage symptoms. The authors might also consider including a healthy olive by the side to improve it.
Answer: this is added now in line 153.
L 135-136. Family mention can be removed here since it was already informed in Table 2.
Answer: this is added now in table 2.
L 138. The caption is incomplete.
Answer: this is added now
L 162-167. Figure legends are in the incorrect order and also incomplete. Plus, as there were zero counts in some stages, it can be removed to improve figure presentation.
L 187-190. It appears that temperature might do not affect species abundance. But as I said previously, an analysis is necessary to do so.
Answer: this is added now in line 264-269.
L 212-214. The figure is unnecessary as the authors can include the information in the text.
Answer: this is corrected now.
L 224. Species presence was familiar under 20 °C (Figure 8); there was only in peak when the temperature reached 25 °C in 2019.
Answer: thrips were abundant when temperature compromises between 25°C and 30°C.
Round 2
Reviewer 1 Report
This manuscript has undergone extensive revision and all previous comments and concerns have been adequately addressed. There are a few areas where the English is awkward, but acceptable. The revisions have greatly improved the manuscript.
Author Response
The English has been checked and edited by my colleague.
Reviewer 2 Report
General comments:
My previous concerns regarding evidence that the highlighted species are responsible for the damage seen have been largely addressed. I would add some sort of statement to the results (end of 3.3) that clarifies the damage and provides results from the flower dissections. An example: Given their peak abundances coinciding with flowering and thrips presence in dissected flowers, H. tritici and F. occidentalis were responsible for some of the damage later observed on fruit, as damage was observed on ovaries from dissected flowers. Neohydatothrips amygdali likely contributed to additional damage as the ovaries matured into fruit due to thrips presence during the phenological stages of fruit development and maturation.
The necessary details were added to the methods.
My remaining concern is that there are still numerous syntactical and grammatical errors that need to be corrected before this is acceptable for publication.
Specific comments:
Suggested title revision: “A preliminary survey in an olive grove from Biskra (southeast Algeria) reveals a high diversity of thrips and new records on olive”
Lines 54-66 should be one paragraph.
Lines 95-100 should be revised and consolidated to avoid repetition and clarify the number of flower subsamples.
Lines 202-204 should be in the statistics paragraph of the methods section.
Line 217: If P<0.05, why is it reported as not significant?
Line 262: “The regression analysis of N. amygdali was determined” can be deleted.
Line 263: Was the regression significant? Please also report the p-value.
Lines 268-270: This paragraph seems out of place. I suggest adding it to the paragraph ending at line 144.
Line 284: The font size increases here. Reduce it to be consistent with the body text throughout the manuscript.
Author Response
My previous concerns regarding evidence that the highlighted species are responsible for the damage seen have been largely addressed. I would add some sort of statement to the results (end of 3.3) that clarifies the damage and provides results from the flower dissections. An example: Given their peak abundances coinciding with flowering and thrips presence in dissected flowers, H. tritici and F. occidentalis were responsible for some of the damage later observed on fruit, as damage was observed on ovaries from dissected flowers. Neohydatothrips amygdali likely contributed to additional damage as the ovaries matured into fruit due to thrips presence during the phenological stages of fruit development and maturation.
Answer: this added now. Line 216-221.
The necessary details were added to the methods.
My remaining concern is that there are still numerous syntactical and grammatical errors that need to be corrected before this is acceptable for publication.
Answer: The manuscript was checked by an english scientist (Dr. Laurence Mound)
Specific comments:
Suggested title revision: “A preliminary survey in an olive grove from Biskra (southeast Algeria) reveals a high diversity of thrips and new records on olive”
Answer: This is ajusted now.
Lines 54-66 should be one paragraph.
Answer: This is added now.Line 54-65.
Lines 95-100 should be revised and consolidated to avoid repetition and clarify the number of flower subsamples.
Answer: This is added now. Line: 94-96.
Lines 202-204 should be in the statistics paragraph of the methods section.
Answer: this is added now. Line 119-121.
Line 217: If P<0.05, why is it reported as not significant?
Answer: p-value = 0.25, so P>0.05. This corrected now. Line 211.
Line 262: “The regression analysis of N. amygdali was determined” can be deleted.
Answer: This is deleted now.
Line 263: Was the regression significant? Please also report the p-value.
Answer: Figure 9 clearly shows that N. amygdaly was present on olives only when temperatures were between 20°C and 30°C. The use of linear regression to analyze the data was inappropriate. Inclusion of temperatures below those that allow development and above those that are lethal ensure that the relationship would not be linear, especially since a series of temperature values are included when no thrips are present. So after reflection we preferred to delete this statistical analysis
Lines 268-270: This paragraph seems out of place. I suggest adding it to the paragraph ending at line 144.
Answer: It is done
Line 284: The font size increases here. Reduce it to be consistent with the body text throughout the manuscript.
Answer: This is corrected now.